# ADR-DQPU: A Novel ADR Signal Detection Using Deep Reinforcement and Positive-Unlabeled Learning

Chun-Kit Chung
Dept. of Computer Science and Information Engineering
National University of Kaohsiung
Kaohsiung City, Taiwan
isu10503301A@cloud.isu.edu.tw

Wen-Yang Lin
Dept. of Computer Science and Information Engineering
National University of Kaohsiung
Kaohsiung City, Taiwan
wylin@nuk.edu.tw

*Abstract*—The medical community has grappled with the challenge of analysis and early detection of severe and unknown adverse drug reactions (ADRs) from Spontaneous Reporting Systems (SRSs) like the FDA Adverse Event Reporting System (FAERS), which often lack professional verification and have inherent uncertainties. These limitations have exacerbated the difficulty of training a robust machine-learning model for detecting ADR signals from SRSs. A solution is to use some authoritative knowledge bases of ADRs, such as SIDER and BioSNAP, which contain limited confirmed ADR relationships (positive), resulting in a relatively small training set compared to the substantial amount of unknown data (unlabeled). This paper proposes a novel ADR signal detection method, ADR-DQPU, to alleviate the issues above by integrating deep reinforcement Q-learning and positive-unlabeled learning. Upon validation using FAERS data, our model outperformed six traditional methods, exhibiting an overall accuracy improvement of 26.45%, an average accuracy improvement of 52.15%, a precision enhancement of 1.89%, a recall improvement of 18.57%, and an F1 score improvement of 10.95%. In comparison to two state-of-the-art machine learning methods, our approach demonstrated an overall accuracy improvement of 64.1%, an average accuracy improvement of 28.23%, a slight decrease of 1.91% in precision, a recall improvement of 55.56%, and an F1 score improvement of 45.53%.

*Keywords—ADR signal detection, deep reinforcement learning, deep Q network, ensemble learning, PU-learning, spontaneous reporting system*

## I. INTRODUCTION

Adverse drug reactions (ADRs) are a significant global public health issue [1]. ADR can result in patient hospitalization or prolonged hospital stays, permanent disabilities, permanent injuries, congenital malformations in fetuses and infants, life-threatening situations, and even death. However, ADR can only be discovered through continuous clinical trials. Before a drug is launched, the manufacturer can only study ADRs based on data from a small group of trial participants. This results in some ADRs being undetected during the premarket clinical trials. Therefore, various countries have established their spontaneous reporting systems (SRSs) for adverse drug reaction events, such as the FDA Adverse Event Reporting System (FAERS) [36], collecting data on domestic adverse drug reaction events to promote research related to ADR [1][12][23]. The goal is to detect hidden ADRs as early as possible after a drug is launched and reduce drug risks [34].

In the past few decades, researchers have developed statistical methods such as reporting odds ratio (ROR) [24], proportional reporting ratio (PRR) [13], and Bayesian Confidence Propagation Neural Network (BCPNN) [5] to analyze ADRs provided by SRS. Although these statistical methods have achieved some effectiveness in ADR analysis, their results are not outstanding. They can only identify a small portion of ADRs in SRS's vast data. On the other hand, machine learning has demonstrated excellent performance in data classification, and it can now assist doctors in making medical diagnoses. Research has also shown that deep learning and machine learning can be used for ADR signal detection with promising results [3][6][9], enabling the prompt identification of ADR signals after a drug is released on the market.

A critical and highly overlooked characteristic of SRS is its inferior data quality; for example, the labeling of ADRs has high uncertainty. Even though researchers have endeavored to examine the adverse reaction events provided by the SRS through various methods to confirm the presence of adverse drug reactions associated with the drugs of interest, it remains unknown whether a drug can cause other unlisted ADRs in SRS data. This phenomenon aligns with Positive and Unlabeled learning (PU-Learning) [6]. The data consists of a small number of affirmative labeled cases and many unknown cases. This directly challenges applying supervised learning methods, such as deep learning, to SRS data. In this study, we utilized some ADR knowledge bases, such as SIDER [37] and Stanford Biomedical Network Dataset Collection (BioSNAP) [35] in ADR labeling, obtaining a small amount of positive labeled data, and employed deep reinforcement learning plus the concept of PU-learning to devise a novel ADR signal detection method in the way of self-learning to explore the certainty of unknown data. We performed experiments to evaluate the usability and accuracy of the proposed method using real-life data from the FAERS database. Our method outperforms other state-of-the-art (SOTA) methods, including six standard statistical methods and two machine learning methods.

The remainder of this paper is organized as follows. Section II presents some background knowledge about this work, including ADR signal detection, deep reinforcement learning, PU-learning, and related work, focusing on machine learning or deep learning based methods for ADR signal detection. Our proposed ADR-DQPU method is described in Section III.

XXX-X-XXXX-XXXX-X/XX/$XX.00 ©20XX IEEE

Section IV presents the experiment we conducted on the FAERS data, showing the performance evaluation of our method against SOTAs. Finally, conclusions and some future avenues are provided in Section V.

## II. Background Knowledge and Related Work

### A. Traditional Methods for ADR Signal Detection

Detecting adverse drug reactions involves evaluating the causal connection between drugs and adverse reactions to identify potential ADR signals for further assessment. The commonly used methods for ADR signal detection are based on disproportionality measures, which compare the observed events of interest in a database with the expected events or the imbalance compared to other events. These methods can be categorized into two main types: frequency-based methods and Bayesian methods, both of which were considered in this study. The standard ADR signal detection methods include ROR [24], PRR [13], BPCNN [4][5], Sequential Probability Ratio Test (SPRT) [14], MHRA [30], and Yule's Q [30]. These methods calculate the likelihood of a particular drug $D$ causing some specific adverse reaction $R$ based on the number of cases of adverse events, expressed in the form of a contingency table shown in Table I.

TABLE I.     Contingency table of drug-adverse reaction events

|  | Reaction $R$ | Other Reactions ($\overline{R}$) | Total |
|---|---|---|---|
| *Drug* (*D*) | $a$ | $b$ | $a+b$ |
| Other Drugs ($\overline{D}$) | $c$ | $d$ | $c+d$ |
| Total | $a+c$ | $b+d$ | $a+b+c+d$ |

### B. SIDER and BioSNAP

SIDER is a drug adverse reaction database established by Kuhn *et al.* [16]. The latest version is SIDER 4.1 [40]. A total of 139,756 combinations of drug-adverse reaction relationships are provided, encompassing 1,430 distinct drugs and 5,868 adverse reactions.

BioSNAP is a subproject of the SNAP project [35]. This dataset primarily records data in the form of entity relationships, focusing on biomedical-related information and the relationships between the data. In this study, we utilized the "Drug side-effect association network" in BioSNAP, which represents the associations between drugs available in the US and adverse reactions.

### C. Deep Reinforcement Learning

Reinforcement learning is a machine learning method in which we need an agent to learn from the environment by interacting with it [26]. Like we learn how to ride a bicycle, the agent learns by trial and error, interacting with the environment, receiving feedback, and then making a decision (action). Feedback includes the reward defined by the developer and the next state of the environment. In contrast to supervised learning, the agent of reinforcement learning learns through experimentation rather than imitation. No manual labeling of training data is required. Instead, positive and negative experiences are obtained by interacting and observing the results, which serve as training data.

Q-Learning [32] is an algorithm used in value-based reinforcement learning. It operates based on the Q-value, denoted as Q($s$, $a$), which represents the expected benefit of taking an action $a$ (where $a \in A = \{a_1, a_2, …, a_n\}$) in state $s$ (where $s \in S = \{s_1, s_2, …, s_m\}$) at a specific moment. The environment provides feedback based on the agent's action as a corresponding reward, denoted as $r$. The fundamental concept of the Q-learning algorithm involves constructing a Q-table that stores the Q-values for each state-action pair and the strategy to select the action that yields the maximum benefit based on the corresponding Q-value.

Deep reinforcement learning is a machine learning method that combines techniques from deep learning and reinforcement learning [21][22][28][29]. The main distinguished feature of deep reinforcement learning is that, instead of a Q-function, a neural network is treated as an agent that interacts with the environment.

### D. Positive and Unlabeled Learning

In practical scenarios, instances often arise where a small number of positive labels coexist with a substantial volume of unlabeled data [7]. Take ADR signal detection as an example, where experts can employ various methods to inspect adverse event data from the SRS dataset, confirming the existence of adverse reactions linked to specific drugs of interest. However, it remains uncertain whether a drug could potentially induce other unreported adverse drug reactions within the SRS dataset. Treating unlabeled data directly as negative cases during classifier training in such cases can lead to significant errors [18]. PU-Learning [6][10][15][25] offers a tailored solution for this data scenario. It leverages information from positive cases to analyze unlabeled cases, determining the probabilities of these cases belonging to positive or negative categories. This process aids in identifying new positive (or negative) cases with high probabilities, increasing the amount of training data and mitigating the risk of misclassifying positive instances in the unlabeled data as negatives.

### E. Related Work

In recent years, several researchers have applied machine learning and deep learning techniques for ADR signal detection, achieving promising results [15][27][34].

A scoping review of papers published from 2000 to 2021 by Kompa *et al.* [15] revealed that although traditional statistical-based ADR signal detection methods remained popular, machine learning and deep learning methods have significantly increased since 2015. SVM, Logical Regression, eXtreme Gradient Boosting (XGBoost), and CNN are the most popular among these methods. Another review covering papers from 1998 to 2020 conducted by Syrowatka *et al.* [27] exhibited similar results: Machine learning methods have been widely used in ADR detection since 2015. Below we highlight some of the notable research after 2021.

Wang and Lin [31] proposed a deep learning-based ADR signal detection method using SIDER to filter FAERS data. A CNN model was used to identify whether drug combinations in ADR signal detection exhibit ADR signals. It leverages deep learning to learn more underlying features from FAERS data, resulting in excellent performance in ADR signal detection

tasks. However, this method is trained only for a single adverse reaction. If ADR signal detection is required for other adverse reactions, training a model for each adverse reaction is necessary, resulting in tremendous models and significant computation costs.

Bae *et al.* [3] evaluated two well-known machine learning methods, Gradient Boosting Machine (GBM) and Random Forest against traditional methods for ADR signal detection, ROR and IC, specifically targeting raw FAERS data. Their findings showed that GBM achieved the best average predictive performance. However, this study only used two drugs, Nivolumab and Docetaxel, as examples. Whether GBM can yield similar performance with other drug data remains unknown. A similar but more comprehensive study considering six medicines was conducted by Imran *et al.* [20]. Four machine learning methods were compared: Random Forest, Linear Support Vector Classifier, Logistic Regression, and XGBoost. The results showed that XGBoost exhibited the best performance over the other methods.

Lin and Tseng [18] first observed the PU characteristic of SRS data and proposed an ADR signal detection method based on stacking ensemble learning and PU-learning. The proposed method consists of two steps. Firstly, the stacking approach is employed to perform signal detection based on disproportionality analysis using traditional methods of calculating the statistical values of the data. The results are then organized into new training data for a meta-classifier. This approach combines the advantages of different traditional methods, training a solid classifier that becomes a valuable learning method. However, this method overlooks the potential features of the FAERS data, such as the inclusion of ATC codes, and only utilizes the statistical values of the data.

Attayeb *et al.* [2] proposed a deep learning-based method for ADR signal detection. In addition to FAERS, their method also incorporates drug-induced gene expression profiles from Open TG–GATEs [39]. Consequently, their model can learn and identify ADR signals based on more features, enabling more in-depth detection of adverse drug reaction signals hidden within the data.

Du *et al.* [11] considered the problem of extracting adverse events by deep learning from reports in the Vaccine Adverse Event Reporting System (VAERS) [41], another SRS system. Their findings conclude that deep learning models such as BioBERT and the proposed VAERS BERT outperform traditional machine learning methods. However, their work belongs to named entity recognition (NER) rather than detecting ADR signals.

## III. THE PROPOSED METHOD

### A. Framework Overview

Fig. 1 illustrates the framework of the proposed ADR-DQPU method. Instead of utilizing raw data from FAERS, we leveraged ADR data cubes generated by the iADRs system [19][37]. Each record within these cubes contains statistical contingency values for drug-adverse reaction pairs, reducing noise and uncertainty in the original FAERS data. In the ADR-DQPU process, we initially established a background

knowledge of ADRs using SIDER and BioSNAP, capturing associations between drugs and adverse reactions. This background knowledge was then employed to filter the source data from ADR cubes, labeling positive and unknown data to create a PU dataset for subsequent model training and testing. The PU dataset was partitioned to generate distinct training and testing sets. To address the class imbalance, we adopted undersampling to balance the training set before training the DQN model for ADR signal detection. Finally, the model's performance was evaluated on the testing set, completing the entire process.

### B. Data Preprocessing

As mentioned previously, FAERS data contain false positives, repetitive reporting, and other uncertainty issues. Therefore, we utilized the ADR contingency cubes generated by the iADRs analysis system [19][37] as inputs for our framework. More precisely, we utilized subcubes encompassing drug ATC codes, symptom names, and four corresponding contingency values ($a$, $b$, $c$, $d$) representing the records in the contingency table for ADR signal detection. Since drug ATC code contains various features related to anatomical therapeutic chemical classification, we performed label encoding to convert the English alphabet to numbers. Then we applied one-hot encoding to separate the features within the ATC code. Table II illustrates an example of these subcubes represented in a tabular format. Furthermore, we observed inconsistencies in the reaction names within the dataset, such as differences in capitalization and the hierarchical structure. To address this issue, we encoded the reaction names to the Preferred Term (PT) level using MedDRA [37].

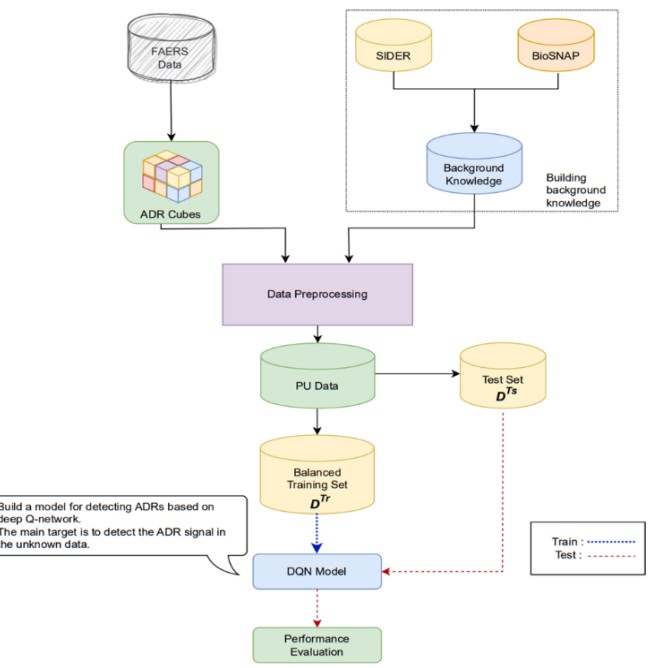

Fig. 1. The framework of the proposed ADR-DQPU method.

TABLE II.    TABULAR REPRESENTATION OF AN EXAMPLE ADR
CONTINGENCY CUBE

| Drug ATC | Symptom | a | b | c | d |
|----------|---------|---|---|---|---|
| C03EA01 | ASCITES | 2 | 1361 | 28 | 19642 |
| R06AE09 | MYALGIA | 6 | 351 | 298 | 35532 |
| M03BX08 | BRONCHITIS | 4 | 451 | 312 | 35420 |
| N06AX23 | ARTHROPATHY | 2 | 233 | 22 | 15105 |
| G04CA02 | BLOOD COUNT ABNORMAL | 1 | 133 | 2 | 6069 |

## C. Design of the DQN Model

The kernel of our proposed ADR-DQPU framework is a deep Q learning-based model (DQN) for ADR signal detection. An architectural flow diagram of the method is illustrated in Fig. 2.

The proposed model follows the basic DQN structure through the interaction between a neural network-based agent and the environment to train the DQN-agent to make a good decision for the incoming event. A state $s^t$ corresponds to a record in the data $D^{Tr}$ (as depicted in Table II) and action $a^t$ represents the decision label made by the agent ("1" representing positive, while "0" denoting unknown). Specifically, the DQN-agent's primary role is to predict the maximum expected reward when taking different actions in the current state, that is, to calculate the Q-value for each $(s, a)$ pair.

In every epoch, we first reset the environment. Initially, the environment sends a random state $s^t$ from PU data $D^{Tr}$ to DQN-agent; the agent detects the state $st$ and provides an action $at$ and returns to the environment. This means the agent decides whether an ADR signal should be generated based on the state $s^t$. On all subsequent steps, the environment will send a new random state $s^{t+1}$ and a reward $r^t$ based on the received action.

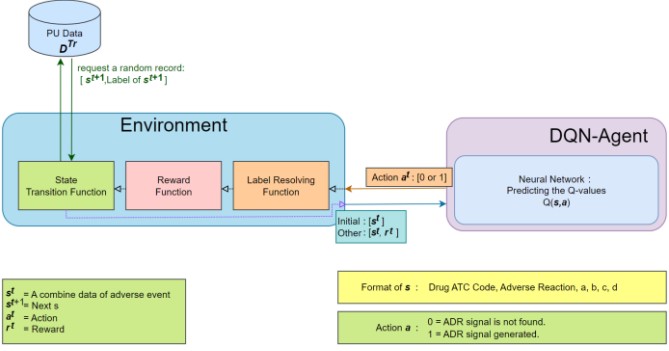

Fig. 2. The architectural flow diagram of the DQN model.

We adopted the well-known double Q-network [29], which is composed of two networks, Q-network and target Q-network. When the DQN-agent receives a state $s^t$, the Q-network will predict the Q-value of each action in this state $s^t$, and then invoke the $\varepsilon$-greedy policy to determine whether to output a random action $a^t$ or the action $a^t$ with the highest Q-value as predicted by the Q-network. The DQN-agent stores all this information in an experience replay buffer in the format of $(s^t, a^t, r^t, s^{t+1})$ as transition data, and then randomly sample a transition data to train the target Q-network to calculate the

target Q-value. We incorporate the target Q-value into the loss function to calculate the mean-square error, enabling the Q-network to perform gradient descent-based backpropagation. Once we have trained for a sufficient number of mini-batches, we duplicate the weights of the Q-network to update the weights of the target Q-network. The agent will run until the environment signals the end of an episode with a 'done' state. The organization and workflow of the DQN-agent is shown in Fig. 3.

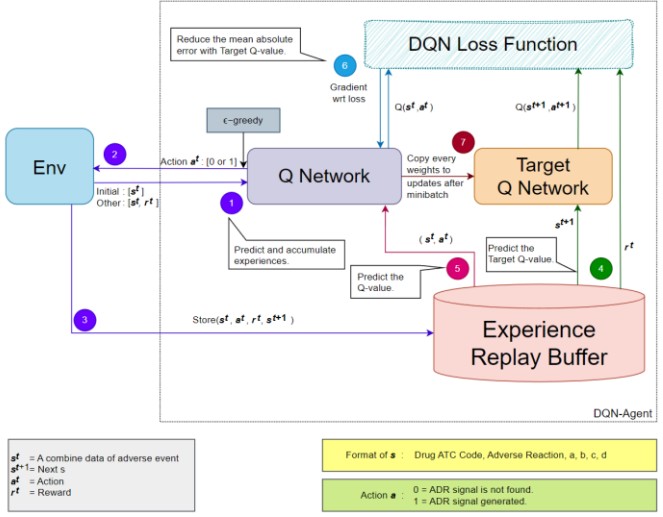

Fig. 3. The organization and workflow of the DQN-agent.

## D. Environment Building

The reinforcement learning environment consists of three main components: state transition function, label-resolving function, and reward function. The primary purpose of the reinforcement learning environment is to send a state $s^t$ (ADR combination data) from PU data $D^{Tr}$ to DQN-agent and then receive action $a^t$ return from the DQN-agent to calculate the reward $r^t$, and send it with the next state. Fig. 4 depicts an architectural diagram of the environment.

When the environment receives an action $a^t$, it will first call the label-resolving mechanism. If the corresponding state $s^t$ belongs to unknown data, the function will determine whether it needs to be relabeled as '1' using an ensemble ADR detection method [18], including ROR, PRR, MHRA, BCPNN, SPRT, and Yule's Q, each performing ADR signal detection on the drug-adverse reaction combination data. Then, a weighted voting approach is used to determine whether the data can be confirmed as a positive case, labeled as '1', or retained in the label '0' as an unknown case.

The reward function is designed to calculate the reward (or penalty) based on the data label and the ADR signal detection result (action $a^t$) produced by the DQN-agent. The formula for computing reward $r$ is as follows:

$$r = \begin{cases} 1, & \text{if } a^t = s^t.Label, \\ -1, & \text{otherwise.} \end{cases}$$

where a reward of 1 is given if the label is positive ('1') and the model correctly detects an ADR signal ('1'), or the label is unknown ('0') and the model correctly detects no ADR signal ('0'); otherwise, a penalty of −1 is given.

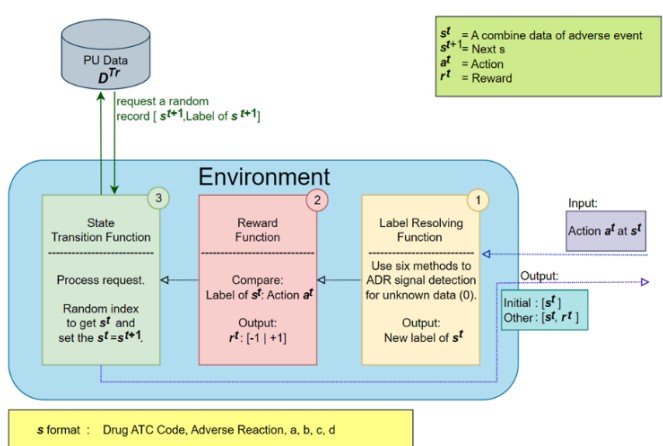

Fig. 4. The architecture of the environment.

## IV. EMPIRICAL EVALUATION

### A. Experimental Design

*a) Data Source and Preprocessing:* FAERS accepts reports from anyone without verification, so the collected data may contain incorrect information. For example, some fields like age and gender may be left blank by the reporters, or adverse reactions caused by other factors may be attributed to drug intake. To address this issue, we utilized the contingency ADR cubes available in iADRs [37], which reduces the significance of individual data points through statistical analysis. We used the contingency cubes from iADRs, covering the first quarter of 2004 to the first quarter of 2019. The datasets from 2004 to 2018 were used as the training data, and the first quarter of 2019 was used as the test data. This choice was made because FAERS has been widely utilized for adverse reaction event reporting, and the data collected within a quarter is sufficient for testing purposes. Labels that indicate whether a drug-adverse reaction combination produces a signal were determined by accessing information provided by the two knowledge systems, SIDER and BioSNAP. In total, we identified 301,644 drug-adverse reaction relationships. After filtering through the background knowledge constructed from SIDER and BioSNAP, we obtained 65,489,379 records of drug-adverse reaction combinations, as shown in Table III. It is noteworthy that the data are highly imbalanced (potitive/unlabeled ratio = 0.076). We thus performed undersampling on the training set to balance the data, reducing the amount of unknown data (label = 0) to achieve an equal number of instances for label = 1 and label = 0, thereby creating a balanced training dataset. Except for the traditional statistics-based methods, each experimental method was run on the same undersampled dataset, including our proposed method and two machine learning methods.

TABLE III. STATISTICS OF THE EXPERIMENTAL DATA

| Training set (2004-2018) | | Testing set (2019 Q1) | |
|---|---|---|---|
| Positive | Unknown | Positive | Unknown |
| 4,338,627 | 57,048,523 | 261,137 | 3,841,092 |
| **Amount of Training set** | | **Amount of Testing set** | |
| 61,387,150 | | 4,102,229 | |

*b) Experimental Environment:* All experiments were conducted on a PC workstation with the following specifications: an Intel Core i7-8700 processor, 32GB RAM, a 1TB hard disk, and an NVIDIA GeForce RTX 2060 6GB graphics card. The workstation was running on the Windows 10 operating system. All programs were implemented using Python 3.7, and the following packages were utilized: TensorFlow-GPU 2.10.0, Keras 2.10.0, Keras-RL2 1.0.5, Gym 0.18.0, mpmath 1.2.1, pandas 1.2.4, and NumPy 1.21.6.

*c) Competitor Methods:* Besides the six traditional ADR detection methods mentioned in Section II.A, we compared our method with two other machine learning-based methods for ADR signal detection. They are the ADR signal detection method using ensemble learning and PU-learning [18] and the XGBoost method based on gradient boosting [20]. The parameter setting for each method is shown in Table IV. The criterion for the six statistical detections to issue an ADR signal followed the literature [12][30]. We used the Adam optimizer to adjust the hyperparameters dynamically. After testing various initial learning rates, we found that an initial rate of 3e-4 yielded the best results. We also experimented with different exploration policies during training our model, including the E-Greedy Q Policy, Boltzmann Q Policy, and Max Boltzmann Q Policy. Our performance comparisons showed that the E-Greedy policy outperformed the other two strategies in this task

*d) Performance Metrics:* We chose six metrics to evaluate our experiments, including accuracy, weighted accuracy, weighted precision, weighted recall, and weighted f-measure. We adopted the weighted average evaluation metric [8] rather than the commonly used micro average owing to the significant class imbalance in our PU data; over 90% of testing data are unknown.

TABLE IV. PARAMETER SETTINGS FOR ADR METHODS IN THIS STUDY

| | Criterion |
|---|---|
| **ROR$_{95}$** | 95% $CI$ limit > 1 |
| **PRR$_{95}$** | 95% $CI$ limit > 1, $a \geq 3$ |
| **MHRA** | PRR $\geq 2$, $x^2 \geq 4$, $a \geq 3$ |
| **BCPNN** | $E(IC) - 1.96 * SD(IC) > 0$ |
| **SPRT** | $\ln(2) \times a - E(a) \geq 2.94$ |
| **Yule'S Q** | $LI_{95}(Q) > 0$ |
| | **Parameters** |
| **XGBoost** | # of trees: 100, max depth: 50, learning rate: 0.3 |
| **PU-LR** | - |
| **ADR-DQPU** | # of runs: 200000, learning rate: 0.0003 |

## B. Experimental Results

*a) Comparison with SOTAs:* Our method was compared with the two machine learning-based and six traditional methods. Table V displays the results, where values highlighted in bold represent the best result for each metric. From the results, we observe:

- All of the traditional methods exhibit high overall accuracy, precision, recall, and F1-score, but with modest average accuracy, owing to the testing data consisting of a small amount of labeled data and many unlabeled data. Some indicators may appear high due to the use of weighted averaging, which amplifies the values of the scarce labeled data. Therefore, these methods are not particularly effective based on the average accuracy.

- Among the two machine learning methods, XGBoost demonstrates high precision and modest average accuracy but significantly low performance on all other metrics. On the other hand, PU-LR, thanks to embracing PU-learning and ensemble learning, shows much better results in all metrics except average accuracy.

- Our proposed ADR-DQPU method achieves excellent results with only a slightly decreasing precision. These results conclude that our proposed method outperforms other comparative methods in ADR signal detection.

TABLE V. THE PERFORMANCE OF ADR SIGNAL DETECTION METHODS. DATA WAS SOURCED FROM IADRS CUBES (JANUARY - FEBRUARY 2019) AND GENERATED MONTHLY

| | Overall Accuracy | Average Accuracy | Precision | Recall | F1 |
|---|---|---|---|---|---|
| ROR | 0.79 | 0.51 | 0.86 | 0.79 | 0.82 |
| PRR | 0.79 | 0.51 | 0.86 | 0.79 | 0.82 |
| BCPNN | 0.83 | 0.50 | 0.85 | 0.83 | 0.84 |
| MHRA | 0.79 | 0.51 | 0.86 | 0.79 | 0.82 |
| SPRT | 0.88 | 0.51 | 0.86 | 0.88 | 0.87 |
| Yule'S Q | 0.63 | 0.48 | 0.85 | 0.63 | 0.71 |
| XGBoost | 0.09 | 0.50 | **0.93** | 0.09 | 0.02 |
| PU-LR | 0.79 | 0.51 | 0.86 | 0.79 | 0.82 |
| ADR-DQPU | **0.97** | **0.87** | 0.89 | **0.91** | **0.89** |

The iADRs data previously mentioned was derived from monthly records. Recognizing that utilizing data at a finer granularity might yield less prominent ADR signals, we conducted an additional experiment by aggregating the data into quarters. This aimed to enhance ADR signal strength by increasing the statistical value of individual adverse event data. Table VI presents the model evaluation results on quarterly data, revealing similar phenomena.

Surprisingly, the performance of XGBoost trained on quarterly data exhibited significant improvement, with an average accuracy reaching 0.87 and a precision of 0.96. However, the overall accuracy stood at 0.76, along with a moderate recall of 0.7. We suspect that XGBoost struggles to learn ADR signals from monthly data because the differences in the contingency values ($a$, $b$, $c$, and $d$) between label = 1 and label = 0 are not pronounced enough. This, coupled with a small feature set—drug and the four contingency values—limits

XGBoost's ability to generate diverse decision trees. This results in the fact that while XGBoost excels in precision, as evidenced by its cautious approach, its overly conservative nature misses a significant number of ADR signals, as indicated by the recall metric. When the data is aggregated quarterly, though the positive/unlabeled ratio is similar to that of monthly data, the differences in the contingency values become more pronounced; positive cases within label = 1 increase, while contingency values within label = 0 remain relatively stable. This suggests that XGBoost might perform better with coarser granularity, such as yearly aggregation. Nevertheless, since our objective is to detect ADR signals as early as possible, our primary focus remains on learning from monthly data.

TABLE VI. THE PERFORMANCE OF ADR SIGNAL DETECTION METHODS. DATA WAS SOURCED FROM IADRS CUBES (JANUARY - FEBRUARY 2019) AND GENERATED QUARTERLY

| | Overall Accuracy | Average Accuracy | Precision | Recall | F1 |
|---|---|---|---|---|---|
| ROR | 0.70 | 0.62 | 0.92 | 0.70 | 0.79 |
| PRR | 0.70 | 0.62 | 0.92 | 0.70 | 0.79 |
| BCPNN | 0.86 | 0.51 | 0.91 | 0.86 | 0.88 |
| MHRA | 0.76 | 0.46 | 0.90 | 0.76 | 0.82 |
| SPRT | 0.85 | 0.49 | 0.90 | **0.95** | 0.87 |
| Yule'S Q | 0.53 | 0.55 | 0.91 | 0.53 | 0.65 |
| XGBoost | 0.76 | **0.87** | **0.96** | 0.76 | 0.83 |
| PU-LR | 0.70 | 0.60 | 0.92 | 0.70 | 0.79 |
| ADR-DQPU | **0.95** | 0.72 | 0.91 | 0.91 | **0.90** |

Tables VII and VIII provide an overview of the performance improvement achieved by our method compared to each comparator. Columns indicating negative growth are underscored for clarity. Relative to the average performance of six traditional methods, our approach demonstrates a remarkable overall accuracy increase of 26.45%, an average accuracy boost of 52.15%, a slight gain of precision by 1.89%, a recall improvement of 18.57%, and an enhanced F1 score by 10.95%. In comparison to the average performance of the two machine learning methods, our approach yields notable advancements, including a 64.1% increase in overall accuracy, a 28.23% boost in average accuracy, a marginal 1.91% decrease in precision, a substantial 55.56% improvement in recall, and an impressive 45.53% rise in the F1 score.

*b) Ablation Study:* In this experiment, we analyzed the effect of different mechanisms on our proposed ADR-DQPU, including the ensemble label-resolving function based on PU-learning and class balancing using undersampling. Table XI presents the performance results of variants of ADR-DQPU, where ADR-DQN denotes the version without PU-inspired label-resolving mechanism, and ADR-DQPU$_{imb}$ represents that without performing class balancing function. All models were trained on monthly generated data with 20000 runs. The results show that the ensemble label-resolving mechanism significantly increased the identification of positive cases from unknown data and enhanced overall model performance. The class balancing mechanism, which reduces the number of unknown cases in the training set to hinder the model from focusing more on the majority class, also plays an important role in boosting performance.

TABLE VII.  IMPROVEMENT OF OUR METHOD OVER OTHER METHODS ON MONTHLY DATA

| | Overall Accuracy | Average Accuracy | Precision | Recall | F1 |
|---|---|---|---|---|---|
| ROR | 22.8% | 70.6% | 3.5% | 15.2% | 8.5% |
| PRR | 22.8% | 70.6% | 3.5% | 15.2% | 8.5% |
| BCPNN | 16.9% | 74.0% | 4.7% | 9.6% | 6.0% |
| MHRA | 22.8% | 70.6% | 3.5% | 15.2% | 8.5% |
| SPRT | 10.2% | 70.6% | 3.5% | 3.4% | 2.3% |
| Yule'S Q | 54.0% | 81.3% | 4.7% | 44.4% | 25.4% |
| XGBoost | 977.8% | 74.0% | -4.3% | 911.1% | 4350.0% |
| PU-LR | 22.8% | 70.6% | 3.5% | 15.2% | 8.5% |

TABLE VIII.  IMPROVEMENT OF OUR METHOD OVER OTHER METHODS ON QUARTERLY DATA

| | Overall Accuracy | Average Accuracy | Precision | Recall | F1 |
|---|---|---|---|---|---|
| ROR | 35.7% | 16.1% | -1.1% | 30.0% | 13.9% |
| PRR | 35.7% | 16.1% | -1.1% | 30.0% | 13.9% |
| BCPNN | 10.5% | 41.2% | 0.0% | 5.8% | 2.3% |
| MHRA | 25.0% | 56.5% | 1.1% | 19.7% | 9.8% |
| SPRT | 11.8% | 46.9% | 1.1% | -4.2% | 3.5% |
| Yule'S Q | 79.3% | 30.9% | 0.0% | 71.7% | 38.5% |
| XGBoost | 25.0% | -17.2% | -5.2% | 19.7% | 8.4% |
| PU-LR | 35.7% | 20.0% | -1.1% | 30.0% | 13.9% |

TABLE XI.  PERFORMANCE RESULTS OF THE PROPOSED METHOD WITH DIFFERENT MECHANISM ABLATION

| | Overall Accuracy | Average Accuracy | Precision | Recall | F1 |
|---|---|---|---|---|---|
| ADR-DQN | 0.62 | 0.55 | 0.89 | 0.62 | 0.72 |
| ADR-DQPU$_{imb}$ | 0.41 | 0.58 | 0.78 | 0.41 | 0.35 |
| ADR-DQPU | **0.86** | **0.84** | **0.93** | **0.85** | **0.88** |

## V.  CONCLUSIONS

Spontaneous reporting systems, like FAERS, have been globally established for drug safety monitoring and early detection of unknown serious adverse drug reactions. Previous efforts have introduced disproportionality-based ADR signal detection methods using FAERS data. However, these traditional approaches may lack effectiveness and overlook hidden features within reported adverse events. This paper presents a novel ADR signal detection method employing deep reinforcement and PU-learning. A preliminary experiment on the FAERS dataset was conducted to assess our framework's performance. As a case study, we predicted ADR signals between drugs and adverse reactions using FAERS reports from the first quarter of 2019. Our proposed ADR-DQPU[1] method demonstrated superior ADR signal detection. Leveraging various mechanisms, we achieved outstanding performance compared with SOTAs. In the future, we will conduct more experiments to evaluate the effectiveness of the proposed ADR-DQPU method on early ADR signal detection of newly marketed drugs. We also like to investigate this approach using other SRS data, such as VAERS.

This study has several limitations. First, we relied solely on SIDER and BioSNAP to determine the certainty of ADR labels.

[1] Code available at GitHub: https://github.com/NUKCILAB/ADR-DQPU

Including additional authoritative databases could enhance both the certainty and quantity of positive cases, potentially influencing our results. Second, we only used contingency values and drug ATC codes, neglecting demographic features like age, gender, and weight. Incorporating these features could facilitate stratified detection of ADR signals across different patient groups, though it would encounter the challenge of handling many missing values (for example, over 70% of weight in FAERS are missing in the demographic features). Lastly, we focused only on adverse reactions from single drugs, whereas interactions between multiple drugs can also cause adverse reactions. An interesting research question is how to adapt our method to detect drug-drug interactions.

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
