# OpenReview forum: "ADR-DQPU: A Novel ADR Signal Detection Using Deep Reinforcement and Positive-Unlabeled Learning"
_IEEE.org/EMBS/BHI/2024/Conference — IEEE BHI'24_

### Official Review · Reviewer_rRoi · 2024-07-29
**The paper makes a potentially significant contribution to the field of ADR signal detection, introducing a novel method with promising results.**

**Overall Rating:** 7
**Confidence:** 3

**Other Quality Metrics:**

(a) Clarity of writing: great;
(b) Clinical Significance: great;
(c) Methodological Novelty: great;
(d) Experiments and Results: excellent

**Questions For The Authors:**

N/A

**Strengths:**

1. The paper proposes a novel approach that combines deep reinforcement learning and positive-unlabeled learning for ADR signal detection, and tackle specific issues in ADR detection, namely uncertain and imbalanced data, which are significant problems in the field..
2. The paper provides detail implementation and key reference to help understand the proposed methods.
3. The paper leverages various data sources (FAERS, iADRs, SIDER, BioSNAP) to create a more comprehensive dataset for training and evaluation.
4. The paper presents specific performance improvements, giving clear metrics for the method's effectiveness, and compared with six traditional statistical methods and two machine learning methods.

**Summary Of The Paper:**

This paper introduces ADR-DQPU, a method for detecting adverse drug reaction (ADR) signals from spontaneous reporting system data like FAERS. The approach combines deep reinforcement learning, specifically deep Q-learning, with positive-unlabeled (PU) learning to address challenges of uncertain and imbalanced data in ADR detection. ADR-DQPU utilizes ADR contingency cubes from the iADRs system as input and leverages background knowledge from SIDER and BioSNAP databases. The method incorporates a deep Q-network agent, an environment with specialized functions, ensemble ADR detection for relabeling unknown data, and class balancing via undersampling. Experimental results on FAERS data demonstrated that ADR-DQPU outperformed six traditional statistical methods and two machine learning methods, showing significant improvements in overall accuracy, average accuracy, recall, and F1 score. Ablation studies confirmed the importance of the PU-inspired label resolving and class balancing mechanisms. The authors conclude that ADR-DQPU shows promise for more effective ADR signal detection from spontaneous reporting data compared to existing approaches.

**Weaknesses:**

1. There's no mention of extensive hyperparameter tuning or sensitivity analysis, which could affect the robustness of the results.
2. The summary doesn't mention a thorough discussion of the method's limitations, which is crucial for a balanced scientific presentation.

---

### Official Review · Reviewer_9hFN · 2024-08-08
**Reinforcement learning + PU learning to tackle unlabeled data**

**Overall Rating:** 7
**Confidence:** 3

**Other Quality Metrics:**

(a) Clarity of writing: Great
(b) Clinical Significance: Great
(c) Methodological Novelty: Good
(d) Experiments and Results: Great

**Questions For The Authors:**

1. Can you clarify the "quarterly" changes mentioned in section B. Experimental Results? Does the positive/unlabeled ratio differ from monthly? It was not clear how the data would be processed and/or if it is different from monthly. A brief explanation would help readers understand this process since the performance for XGBoost was quite different. Additionally, the authors mention "less prominent ADR signals" with the quarterly data—could this be due to time lags in adverse reactions?
2. Why might XGBoost have such low performance in Table V but then better in Table VI? It seems like it was not learning the positive labels from the monthly data. Could there have been significantly fewer positive cases in the quarterly data?

**Strengths:**

1. The writing is clear and effectively identifies the gap in the existing literature, explaining the rationale behind the model design.
2. The framework demonstrates good performance and is supported by an appropriately designed experiment. I especially appreciated the experiment in Table XI.

**Summary Of The Paper:**

The authors propose a novel framework that leverages deep learning and PU learning to enhance the detection of Adverse Drug Reactions (ADRs). This framework addresses the challenges posed by limited training datasets and ambiguously labeled data through self-learning capabilities.

**Weaknesses:**

1. The paper lacks clarity on the undersampling preprocessing step of the dataset. Specifically, after undersampling, how many samples were retained, and what was the new class ratio?
2. Were the other models, such as XGBoost, also subjected to undersampling? Clarifying this would be important because the performance change of ADR-DQPU_imb to the rest was significant, indicating that the undersampling step played a significant role in performance improvement. It would only be fair to compare with the rest of the competitor methods if the same undersampling was performed.
3. Minor comments: typo in the Ablation study where I think authors were pointing to "ADR-DQN" as the version without PU label-resolving mechanism but the text mentions ADR-DQ.

---

### Official Review · Reviewer_WXyN · 2024-08-13
**Several issues need to be addressed in this study.**

**Overall Rating:** 7
**Confidence:** 4

**Other Quality Metrics:**

N/A

**Questions For The Authors:**

N/A

**Strengths:**

The authors proposed a novel ADR signal detection method by integrating deep reinforcement learning with positive-unlabeled (PU) learning.

**Summary Of The Paper:**

The authors proposed a novel ADR signal detection method by integrating deep reinforcement learning with positive-unlabeled (PU) learning. This approach demonstrates significant novelty and practical applicability. The experiments conducted are thorough, and the analysis of the results is relatively reliable. However, several issues need to be addressed in this study.

**Weaknesses:**

1. The Abstract section is too long. Suggest streamlining it appropriately to highlight the paper’s contribution.
2. In section II. BACKGROUND KNOWLEDGE AND RELATED WORK, the background work section is too long and the related work section is too short. Please rearrange the language to streamline and expand them
3. In TABLE VI, the performance of XGBoost outperforms the proposed on Average Accuracy and Precision, Please give reasons.
4. In the ablation study, the authors only analyzed the effect of different mechanisms, but not analyze the reasons for the results. Please provide it.

---

### Decision · Program_Chairs · 2024-09-23

Accept